# Reversible non-volatile electronic switching in a near-room-temperature van der Waals ferromagnet

Han Wu [1], Lei Chen[1], Paul Malinowski [2], Bo Gyu Jang[3,4], Qinwen Deng [5], Kirsty Scott[6,7,8,9], Jianwei Huang [1], Jacob P. C. Ruff[10], Yu He [11], Xiang Chen[11], Chaowei Hu [2,12], Ziqin Yue[1], Ji Seop Oh[1,12], Xiaokun Teng [1], Yucheng Guo[1], Mason Klemm[1], Chuqiao Shi [13], Yue Shi[2], Chandan Setty[1], Tyler Werner[9], Makoto Hashimoto [14], Donghui Lu [14], Turgut Yilmaz[15], Elio Vescovo[15], Sung-Kwan Mo [16], Alexei Fedorov [16], Jonathan D. Denlinger [16], Yaofeng Xie [1], Bin Gao [1], Junichiro Kono [1,13,17], Pengcheng Dai[1], Yimo Han [13], Xiaodong Xu [2,12], Robert J. Birgeneau [11,18,19], Jian-Xin Zhu [3], Eduardo H. da Silva Neto [6,7,8,9], Liang Wu[5], Jiun-Haw Chu [2], Qimiao Si [1] & Ming Yi [1] ✉

Non-volatile phase-change memory devices utilize local heating to toggle between crystalline and amorphous states with distinct electrical properties. Expanding on this kind of switching to two topologically distinct phases requires controlled non-volatile switching between two crystalline phases with distinct symmetries. Here, we report the observation of reversible and non-volatile switching between two stable and closely related crystal structures, with remarkably distinct electronic structures, in the near-room-temperature van der Waals ferromagnet $Fe_{5-\delta}GeTe_2$. We show that the switching is enabled by the ordering and disordering of Fe site vacancies that results in distinct crystalline symmetries of the two phases, which can be controlled by a thermal annealing and quenching method. The two phases are distinguished by the presence of topological nodal lines due to the preserved global inversion symmetry in the site-disordered phase, flat bands resulting from quantum destructive interference on a bipartite lattice, and broken inversion symmetry in the site-ordered phase.

Materials that can toggle between two states with distinct properties are important for information storage technology. Phase-change materials, for example, have been widely used for rewriteable optical data storage[1–19]. The key advantage is that the two phases are controlled by a non-volatile process, which is realized via a transient laser pulse that locally heats and changes the crystal structure, either resulting in a crystalline state or a quenched amorphous state. Two-dimensional van der Waals (vdW) materials is another class of material family whose properties are highly tunable, such as by electrostatic

doping, optical illumination, or strain[20–40]. They are valued not only for their versatile tunability but also for the low dimensionality that allows exotic properties to arise due to quantum confinement. The advent of the concept of topology adds the potential to realize switching devices that go beyond resistive or optical readouts. As topology is often distinguished by crystalline symmetries, the switching between two topologically distinct states can be realized by tuning knobs that alter symmetries, typically achieved through a structural transition[15]. However, these tuning knobs typically involve modulating temperature,

electrostatic doping, strain, electric fields, or pressure, all of which are difficult to achieve using a non-volatile method[1–3,9,10,15–19,26–41].

Here, we demonstrate the non-volatile reversible switching of two closely related crystal structural phases in the vdW ferromagnet $Fe_5GeTe_2$ via an annealing and quenching procedure. $Fe_5GeTe_2$ belongs to a class of Fe-based metallic vdW ferromagnets that exhibit relatively high Curie temperatures ($T_C = 275$–$330$ K in the bulk limit)[42–54]. Different from other widely studied 2D ferromagnets such as $CrI_3$ and $Cr_2X_2Te_6$ (X = Ge, Si)[22,23], $Fe_5GeTe_2$ is air-stable and metallic, hence has been considered a top candidate for spintronics applications[21–23,47,48]. The two phases share a similar overall crystal structure but differ only in the ordering or disordering of a Fe vacancy site occupation that results in distinct crystalline symmetries. Second harmonic generation (SHG) measurements show the site-disordered phase to exhibit global inversion symmetry while the site-ordered phase breaks inversion symmetry, with intensity differing by a factor of 30. Remarkably, the electronic structures in the two phases are qualitatively distinct, as observed by angle-resolved photoemission spectroscopy (ARPES). From a combination of symmetry analysis and first principle calculations, we also provide an understanding of the key features of the observed electronic structure. In the site-disordered phase, we observe topological nodal lines that are compatible with the preserved global inversion symmetry, while in the site-ordered phase, we observe the lifting of the topological degeneracy due to the broken inversion symmetry as well as flat bands that are compatible with the quantum destructive interference of a bipartite crystalline lattice (BCL) formed by the site-ordered phase. Our work not only demonstrates the exciting potential of using site order in the Fe-based 2D materials as a tuning knob to engineer and control correlated topological phases but also reveals the potential of this class of 2D materials as a type of phase-change materials for non-volatile spintronics, memory or non-linear optical applications.

## Results

### Reversible switching of two distinct electronic structures

$Fe_5GeTe_2$ belongs to a larger class of Fe-based metallic ferromagnets, $Fe_nGeTe_2$ ($n = 3$–$5$)[47,48,52,55,56], and is known to have a unique partially occupied split site[47,48]. The crystal lattice of $Fe_5GeTe_2$ is rhombohedral (space group $R\bar{3}m$, No. 166)[47,48]. The crystal structure consists of an ABC-stacking of the vdW slabs (Fig. 1a). Each slab consists of Fe and Ge sites sandwiched between layers of Te. In addition, each slab consists of three distinct Fe sites, marked as Fe(1), Fe(2), and Fe(3) in Fig. 1a. While Fe(2) and Fe(3) sites are fully occupied, Fe(1) sites are known to be split-sites where for each up–down pair within a single slab, they are either occupied in the up or down site[44,47,48]. This choice of either the up or down site for Fe(1) pushes the Ge sites to also occupy a split site, where the site farther away from the occupied Fe(1) site is preferred. The choice of either occupying the up or down Fe(1) sites can be uncorrelated spatially or form an order depending on the rate at which the crystals are formed from growth[47,48]. In particular, the occupancy of the Fe(1) sites could form an up–down–down (UDD) or down–up–up (DUU) pattern, resulting in a $\sqrt{3} \times \sqrt{3}$ superstructure[44]. This ordered occupancy is favored when the crystals are quenched from above a structural transition identified by previous literature as $T_{HT} = 550$ K while the random distribution is favored with slow cooling[47,48]. For simplicity, we refer to the uncorrelated phase the site-disordered phase, and the ordered phase the site-ordered phase. The ordering of the Fe(1) sites plays a crucial role in modifying the global symmetry of the crystal. In the site-disordered phase, the global inversion symmetry is preserved. This can be seen in Fig. 1a, where the inversion centers of each vdW slab are between the Ge split sites. In the site-ordered phase, the inversion symmetry is broken by the Fe(1) sites[44]. Such symmetry breaking has a profound impact on the electronic structure, and as we will demonstrate, is the key to tunability.

To probe such an effect, we carried out ARPES measurements on crystals that were prepared in the two thermal methods. The measured

Fermi surface (FS) of the slow-cooled crystals (Fig. 1g) and the quenched crystals (Fig. 1h) under the same measurement conditions are drastically different. In particular, the quenched crystals exhibit small pockets at the K points of the Brillouin zone (BZ), which are absent in the slow-cooled crystals. Instead, the slow-cooled crystals exhibit additional large pockets centered at the $\Gamma$ point. As we will show in detail in each of the two subsequent sections, the band dispersions leading to these FSs are significantly different, belonging to distinct topologically non-trivial phases. Before we discuss the electronic structure in depth, we first demonstrate the reversible non-volatile switching of these two phases. To confirm that it is the last thermal cooling step that dictates the electronic phase, we performed the following test (see Supplementary Fig. 8). First, we prepared the crystals by quenching them from above $T_{HT}$ down to room temperature. Then we cut a crystal into halves, annealed a half piece to the metastable phase above $T_{HT}$, and slowly cooled it back down to room temperature while leaving the other half untreated (Fig. 1b). The half pieces are then measured by ARPES. The electronic structure of the two halves is observed to be distinct, with the original quenched half identical to that shown in Fig. 1h, while the annealed and slow-cooled half is identical to that presented in Fig. 1g. We have also checked the reverse process, which is to start with a crystal that was first formed via slow-cooling to room temperature, cut it in half, and annealing one half to above $T_{HT}$ and then quenched in cold water (Fig. 1c). The subsequent ARPES measurement on the two halves again show the contrasting electronic structures, with the re-quenched half showing electronic structure identical to that in Fig. 1h and the original slow-cooled half identical to Fig. 1g. This procedure demonstrates that the key for the distinct electronic structures is the cooling rate in the final thermal treatment from above $T_{HT}$. Hence we have demonstrated that there are two stable phases with drastically distinct electronic structures that can be reversibly switched in a non-volatile method.

### Fe(1) site ordering as the origin for distinct electronic phases

As reported, there are three types of possible variations of the $Fe_{5-\delta}GeTe_2$ single crystals: Fe deficiency ($\delta$), stacking faults, and the formation of the $\sqrt{3} \times \sqrt{3}$ Fe(1) site order[44,47,48]. Since each pair of half crystals used above originates from the same original piece, the above procedure also rules out any difference in Fe deficiency as a potential cause for the difference in the electronic structure. Furthermore, we can also rule out the vdW stacking faults as a possible cause of the distinct electronic structure. From our transmission electron microscopy (TEM) images on the two types of crystals (see Supplementary Fig. 1), we do not observe any regular appearance of stacking faults in either the slow-cooled or quenched crystals. Both crystals exhibit ABC stacking, with occasional stacking faults between the vdW layers. Such rare occurrence cannot constitute a qualitative electronic structure distinction between the two types of crystals.

Therefore we are left with the appearance of the $\sqrt{3} \times \sqrt{3}$ Fe(1) site order as the likely cause of the dichotomy of the electronic structure. First from single crystal X-ray diffraction (XRD) measurements, while the diffraction peaks corresponding to the $\sqrt{3} \times \sqrt{3}$ order are observed in the two types of crystals, their intensity relative to the Bragg peaks is reduced in the slow-cooled samples compared to those measured on a quenched crystal (see Supplementary Fig. 3). This suggests that while both site-disordered and site-ordered regions exist in the slow-cooled crystals, the population of the site-ordered regions is smaller. To further confirm this, we carried out scanning tunneling microscopy (STM) measurements on both quenched and slow-cooled crystals, revealing regions with a $\sqrt{3} \times \sqrt{3}$ superlattice with both UDD and DUU ordering of Fe(1) occupation sites (Fig. 1e, f), consistent with previous STM reports on $Fe_{5-\delta}GeTe_2$[44]. While the field of view (on the order of $1\,\mu m^2$) of our STM measurements on quenched crystals showed only $\sqrt{3} \times \sqrt{3}$ ordered regions, similar measurements on slow-cooled crystals also showed disordered regions without the $\sqrt{3} \times \sqrt{3}$

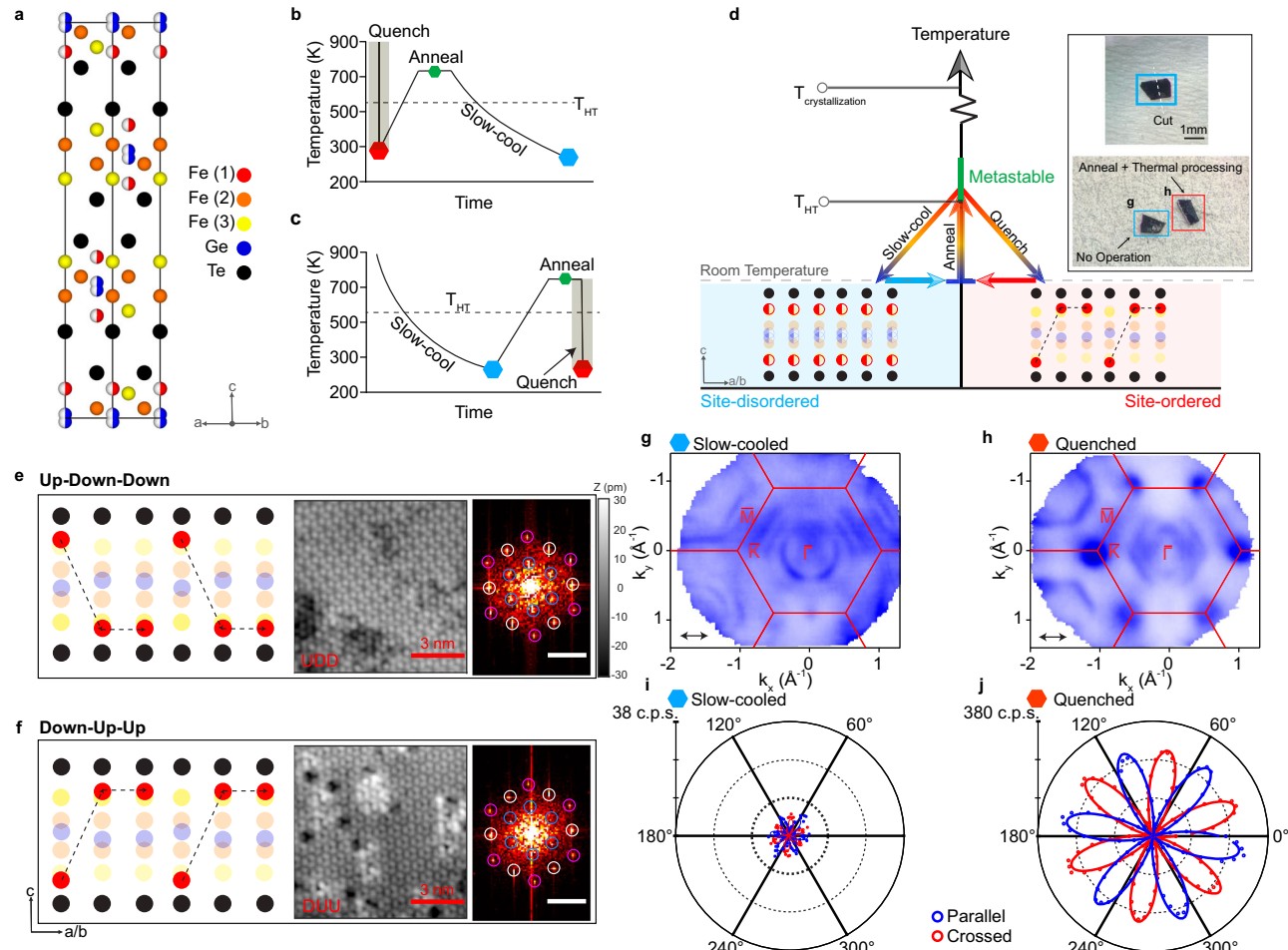

**Fig. 1 | Phase tuning in Fe₅₋δGeTe₂.** **a** Crystal structure of Fe₅GeTe₂ with atomic sites labeled. Fe(1) and Ge are modeled as split sites, marked by half-filled circles to represent 50% occupancy. **b**–**d** Schematic and procedures for tuning global symmetry via sublattice ordering. The two quantum phases can be switched by first annealing to above $T_{HT}$ = 550 K and either quenching or slowly cooling to achieve the site-ordered phase or the site-disordered phase, respectively. The inset in **d** show the real steps of tuning phases. **e**–**f** Schematic model and STM topographic image of the $\sqrt{3} \times \sqrt{3}$ superstructure on Te termination. The right panels are Fourier transforms of the respective topographies with peaks corresponding to the lattice periodicity (white), a $\sqrt{3} \times \sqrt{3}$ superstructure periodicity (blue), and the second order of the $\sqrt{3} \times \sqrt{3}$ superstructure periodicity (pink). The inset scale bars

are 17.3 nm⁻¹. The schematic lattice in **e** corresponds to an up–down–down (UDD) ordering of the Fe(1) atoms while that in **f** corresponds to a down–up–up (DUU) ordering of the Fe(1) sites. **g**–**h** The Fermi surfaces of the two phases were measured at 15 K using 114 eV LV photons, achieved by slow-cooling the crystal to room temperature from above $T_{HT}$ (**g**) and quenching from above $T_{HT}$ (h), respectively. The arrows indicate the photon polarization direction. **i, j** Polarization-resolved SHG intensity on slow-cooled and quenched crystals measured at 5K. In both figures, the crossed and parallel configurations correspond to $E(2\omega) \perp E(\omega)$ and $E(2\omega) \parallel E(\omega)$, respectively, while $E(2\omega)$ and $E(\omega)$ were simultaneously rotated in the crystal $ab$-plane. c.p.s. stands for counts per second. The dots are experimental data and the solid curves are the fits by a sixfold sinusoidal function.

superlattices. Interestingly, in slow-cooled crystals, these disordered regions dominate the field of view, surrounding small domains of $\sqrt{3} \times \sqrt{3}$ superlattices (see Supplementary Fig. 2), consistent with the XRD results. The existence of regions with $\sqrt{3} \times \sqrt{3}$ order in the two types of crystals revealed by STM is further confirmed by SHG measurement. We carried out polarization-dependent SHG measurements at 5K on the two types of crystals. The quenched crystals reveal a 30 times stronger SHG signal compared to that of the slow-cooled crystals (Fig. 1i, j). Note that to observe the tiny SHG signal (less than 10 c.p.s.) in the slow-cooled sample, an incident power of 4 mW with a 50× objective is needed, which is just below the damage threshold, requiring a photon counter. As the SHG signal is contributed by the electric dipole (ED),

$$I_i^{ED}(2\omega) \propto \left| \Sigma_{jk} \chi_{ijk}^{ED} E_j(\omega) E_k(\omega) \right|^2, \tag{1}$$

where $k$ and $\omega$ are the wavevector and frequency of the incident beam, respectively, $\chi$ is the nonlinear susceptibility tensor and $i, j, k, l$ are

Cartesian coordinate indices, it is a sensitive probe of the presence of inversion-symmetry-breaking. On one hand, for the quenched crystals, the clear presence of inversion symmetry breaking is consistent with the formation of the $\sqrt{3} \times \sqrt{3}$ order. On the other hand, in the slow-cooled crystals dominated by regions with random Fe(1) site occupancy, the ED contribution to SHG would be forbidden due to the preserved global inversion symmetry while only a smaller electric quadrupole (EQ) SHG contribution following the threefold rotational symmetry would be allowed,

$$I_i^{EQ}(2\omega) \propto \left| \Sigma_{jkl} \chi_{ijkl}^{EQ} k_j E_k(\omega) E_l(\omega) \right|^2 \tag{2}$$

under normal incidence. The much enhanced SHG signal in the quenched crystals is consistent with both the STM and XRD observations. Hence, we associate the electronic structure measured on the quenched crystals to that of the Fe(1) site-ordered phase and that measured on the slow-cooled crystals to that of the Fe(1) site-disordered phase. As we will demonstrate subsequently in the

discussion section, the crystal symmetries for the site-disordered and site-ordered phases are highly compatible with the topological band dispersions that we observe from ARPES.

### Nodal lines in the site-disordered phase

Next, we present in detail the key features in the measured electronic structure of the site-disordered phase achieved by slow-cooling the crystals. The FS in the ferromagnetically ordered state is shown in Fig. 2a, consisting of several circular Fermi pockets centered at the BZ center and elliptical pockets surrounding the $\bar{K}$–$\bar{M}$–$\bar{K}'$ BZ boundaries, as highlighted by white dashed lines. To understand these features, we measured the electronic dispersions along the high symmetry direction of $\bar{M}$–$\bar{K}$–$\bar{\Gamma}$–$\bar{K}$ (Fig. 2d). First, we observe a number of hole bands centered at the $\bar{\Gamma}$ point, giving rise to the observed circular Fermi pockets. Interestingly, near the $\bar{K}$ point, we also observe a band crossing near −0.18 eV. The crossing can be better visualized from the energy distribution curves (EDC) as well as second energy derivatives of the raw spectra (Fig. 2e). From a three-dimensional zoom in view of the electronic structure around the $\bar{K}$ point (Fig. 2b), we see that the elliptical Fermi pockets are formed by one of the branches that participate in the crossing at K. The details of the crossing can be further demonstrated from the measured band dispersions along a series of parallel cuts in this region near the $\bar{K}$ point. In Fig. 2c, both vertical cuts (cuts 1–5) and horizontal cuts (cuts 6–10) in the crossing region reveal the evolution of the two bands that cross near the $\bar{K}$ point and become gapped away from $\bar{K}$.

Having demonstrated the band crossing at the $\bar{K}$ points in the in-plane direction, we also examine the dispersion along the out-of-plane direction ($k_z$) by varying the photon energy. As the inter-layer interactions in vdW materials are quite weak, we do not observe strong variation along $k_z$ (See Supplementary Fig. 6). For a range of photon energies that probes a range much beyond that of a single BZ along $k_z$, we always observe the crossing near the $\bar{K}$ point (Fig. 2f). Hence the in-plane nodal crossing takes the form of nodal lines along the out-of-plane direction. Taking these findings together, our ARPES data reveal the existence of nodal lines along the BZ boundaries. As these nodal lines are observed in a ferromagnetic phase, the time-reversal symmetry is broken and hence the spin

degree of freedom is quenched, giving rise to twofold degenerate nodal lines.

### Flat bands in the site-ordered phase

Having presented the existence of the nodal lines in the site-disordered phase, we now focus on the observed electronic structure of the site-ordered phase. Figure 3 summarizes the measured electronic structure of quenched crystals. Instead of the elliptical Fermi pockets surrounding the $\bar{K}$–$\bar{M}$ direction resulting from the Dirac crossing at $\bar{K}$ points in the slow-cooled crystals, the quenched crystals exhibit circular pockets at the $\bar{K}$ points. This distinction can be further seen from dispersions measured along the high symmetry direction $\bar{M}$–$\bar{K}$–$\bar{\Gamma}$–$\bar{K}$. In stark contrast to that measured for the site-disordered phase (Fig. 2), the site-ordered crystals show electron bands at the $\bar{K}$ points with clear band bottoms and no band crossings, and hence the absence of the nodal lines observed in the site-disordered crystals.

More interestingly, three flat bands are observed in the site-ordered crystals that are not observed in the site-disordered crystals. We first illustrate them along the $\bar{K}$–$\bar{\Gamma}$–$\bar{K}$–$\bar{M}$ direction, captured in measurements under both linear horizontal (LH) and linear vertical (LV) polarizations (Fig. 3a, b). The location of the flat bands can be identified as peaks in the integrated EDCs from both polarizations, at $E_F$, −0.2 eV and −0.6 eV. Beyond the high symmetry direction, the flat bands are observed to persist across a large region of the BZ. We illustrate this from five cuts measured across the in-plane BZ (Fig. 3d, e). The flat band near $E_F$ could be clearly seen along the $\bar{K}$–$\bar{M}$–$\bar{K}$ direction as shown in cut1. When the $\bar{\Gamma}$ point is approached from cut2 to cut5, the flat band near $E_F$ shifts to above $E_F$ and can no longer be observed. The second and third flat bands located at −0.2 eV and −0.6 eV are flat throughout the BZ except where they hybridize with the dispersive bands near the $\bar{\Gamma}$ point. We note that this hybridization indicates that these flat dispersions are intrinsic to the crystal and cannot be due to disorders or impurities that would otherwise form momentum-independent states that do not interact with intrinsic band structure. Furthermore, we carried out photon energy-dependent measurements, where the flat bands are observed to persist across $k_z$ (see Supplementary Note 5 for details), consistent with the 2D nature of the vdW materials.

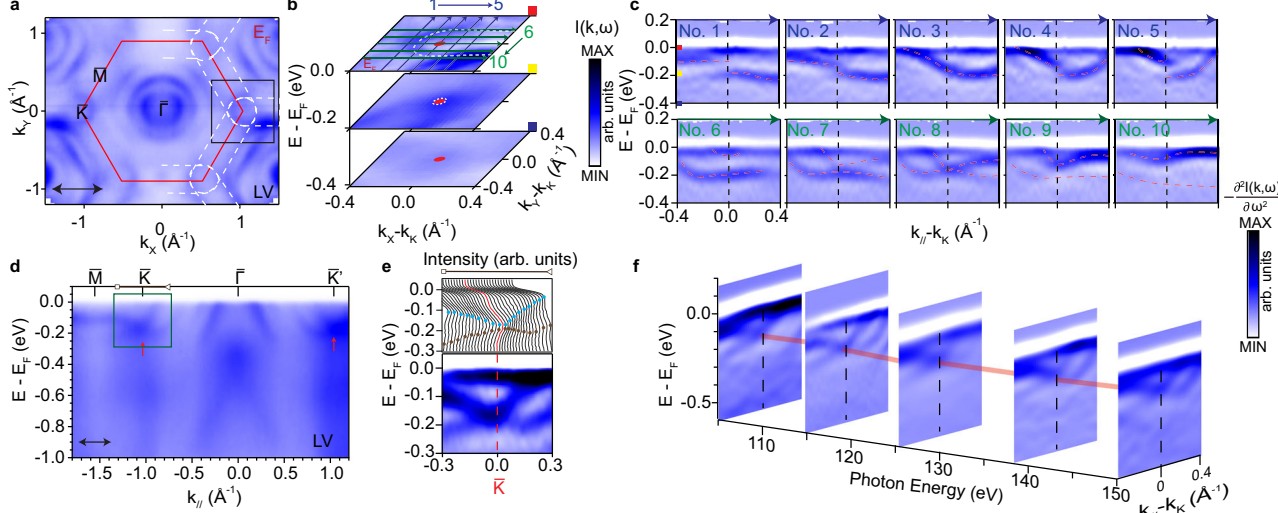

**Fig. 2 | Topological nodal lines in the slow-cooled site-disordered phase.**
**a** Fermi surface mapping with LV polarized photons. The red line outlines the 2D BZ boundary and the white dashed line marks the electron pocket surrounding the BZ boundary. **b** Zoomed in view of the box in (**a**). **c** Dispersions measured across the Dirac node, showing the crossing bands and the opening of a gap away from the Dirac node. **d** Dispersions measured along the $\bar{M}$–$\bar{K}$–$\bar{\Gamma}$–$\bar{K}'$ direction with LV

polarization. The red arrows in (**d**) point to the Dirac crossings. **e** Energy distribution curves (EDCs) and the second energy derivative cut within the momentum range marked by the black arrow in (**d**). **f** Photon energy dependence of the cut near the $\bar{K}$ point. The red solid line shows the Dirac nodal line along $k_z$. All data from **a**–**e** were taken with 132 eV photons.

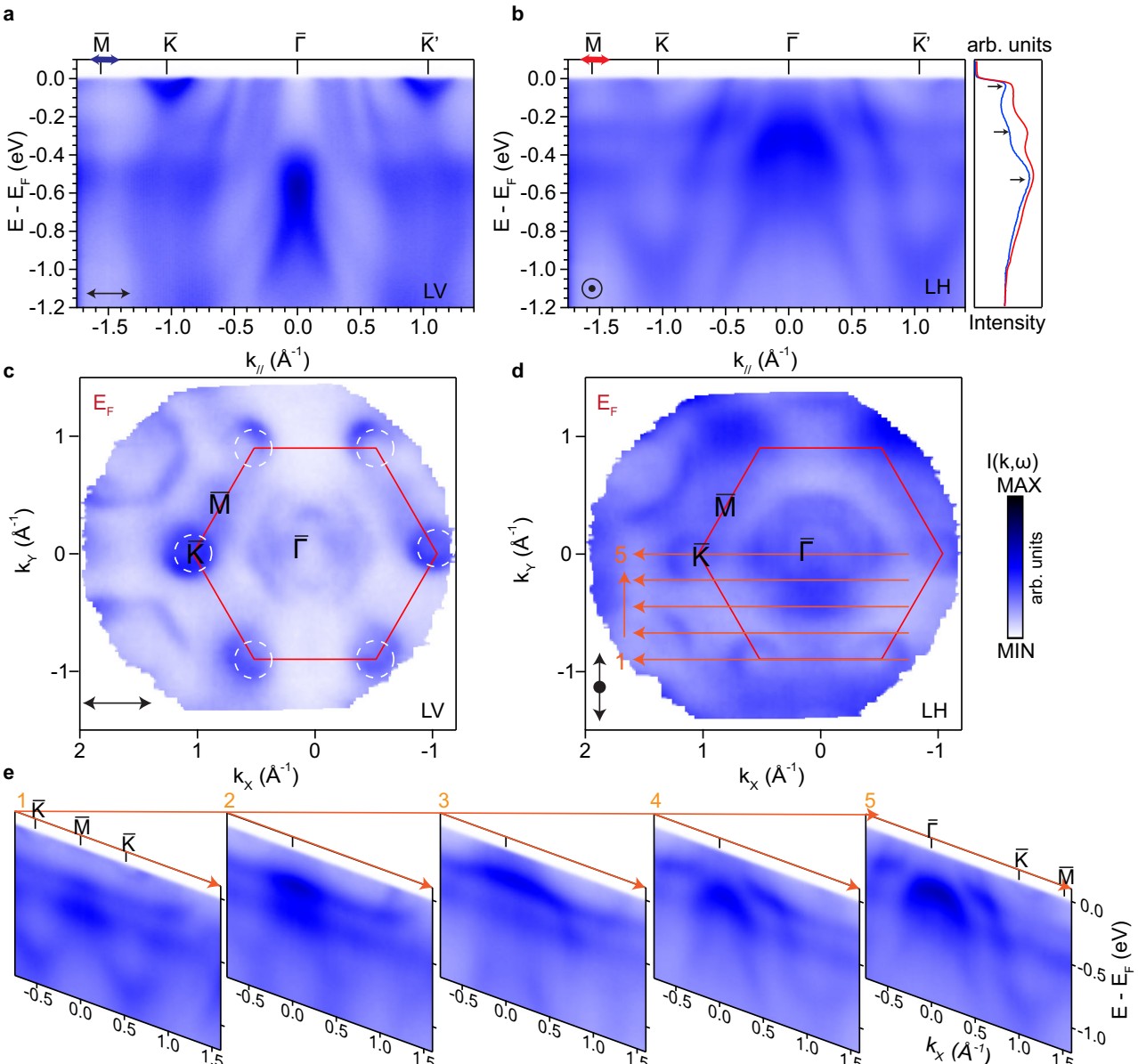

**Fig. 3 | Flat bands in the quenched site-ordered phase. a, b** Measured energy-momentum dispersions along the $\bar{M}-\bar{K}-\bar{\Gamma}-\bar{K}'$ direction with LV and LH polarized photons. The EDCs are integrated from a small range around $\bar{M}$ marked by the arrows in (**a, b**), showing three peaks that correspond to the location of the flat bands. **c, d** Corresponding Fermi surfaces measured with LV and LH polarizations, respectively. Data in **c** were measured under the same geometry as (**b**). **e** Measured dispersions along a series of parallel cuts within the first BZ as shown in (**d**). All data from **a**–**e** were taken with 114 eV photons.

## Topology for the distinct electronic phases

The drastically distinct electronic structures of the two types of crystals, with one exhibiting twofold nodal lines and the other flat dispersions, belong to distinct topological states. Here we show that they can be understood from the symmetries dictated by the site-disordered or site-ordered phases, respectively. We first discuss the case of the site-disordered phase where we observe nodal lines at the K points. For a single vdW slab with 50% occupation of the Fe(1) sites, the crystalline symmetry belongs to the centrosymmetric space group $P\bar{3}1m$ (No. 164). Here, the crystal has both twofold rotational symmetry about the $y$-axis ($C_{2y}$) (Fig. 4b) and threefold rotational symmetry about the $z$-axis ($C_{3z}$) (Fig. 4c), similar to the case of graphene. The momentum point $K$ ($K'$) is invariant under these two symmetry operations and allows the existence of a 2D irreducible representation. In the ferromagnetic phase where time-reversal symmetry is broken, the spin-polarized bands in the ferromagnetic state can be regarded as

spinless states and would cross at the K (K') points, where the twofold degeneracy comes from the orbital degree of freedom (see Supplementary Note 9 for a discussion of the orbitals), leading to a symmetry-enforced ferromagnetic Dirac crossing. To demonstrate this, we built an effective tight-binding model considering the different Fe 3d orbitals and show that such a crossing is orbital-dependent and indeed protected at the K (K') point (see Supplementary Notes 9–10 for a full discussion of the tight-binding model). When we incorporate spin-orbit coupling (SOC), (see Supplementary Note 9), in general, the SOC can be expressed as $H_{SO} = \lambda_{SO}\boldsymbol{L} \cdot \boldsymbol{S}$, where $\boldsymbol{L}$ and $\boldsymbol{S}$ are the angular and spin momenta, respectively. In a ferromagnetic system, $\boldsymbol{S} \cdot \langle \boldsymbol{S} \rangle$ plays the role of an effective Zeeman splitting field in the orbital basis. Since the direction of the magnetic moment is along the $z$ direction[47,48], which is parallel to the direction of the orbital angular momentum, the SOC would lift the twofold degeneracy at K and K'. The band structure calculation from the tight binding model with SOC is illustrated in

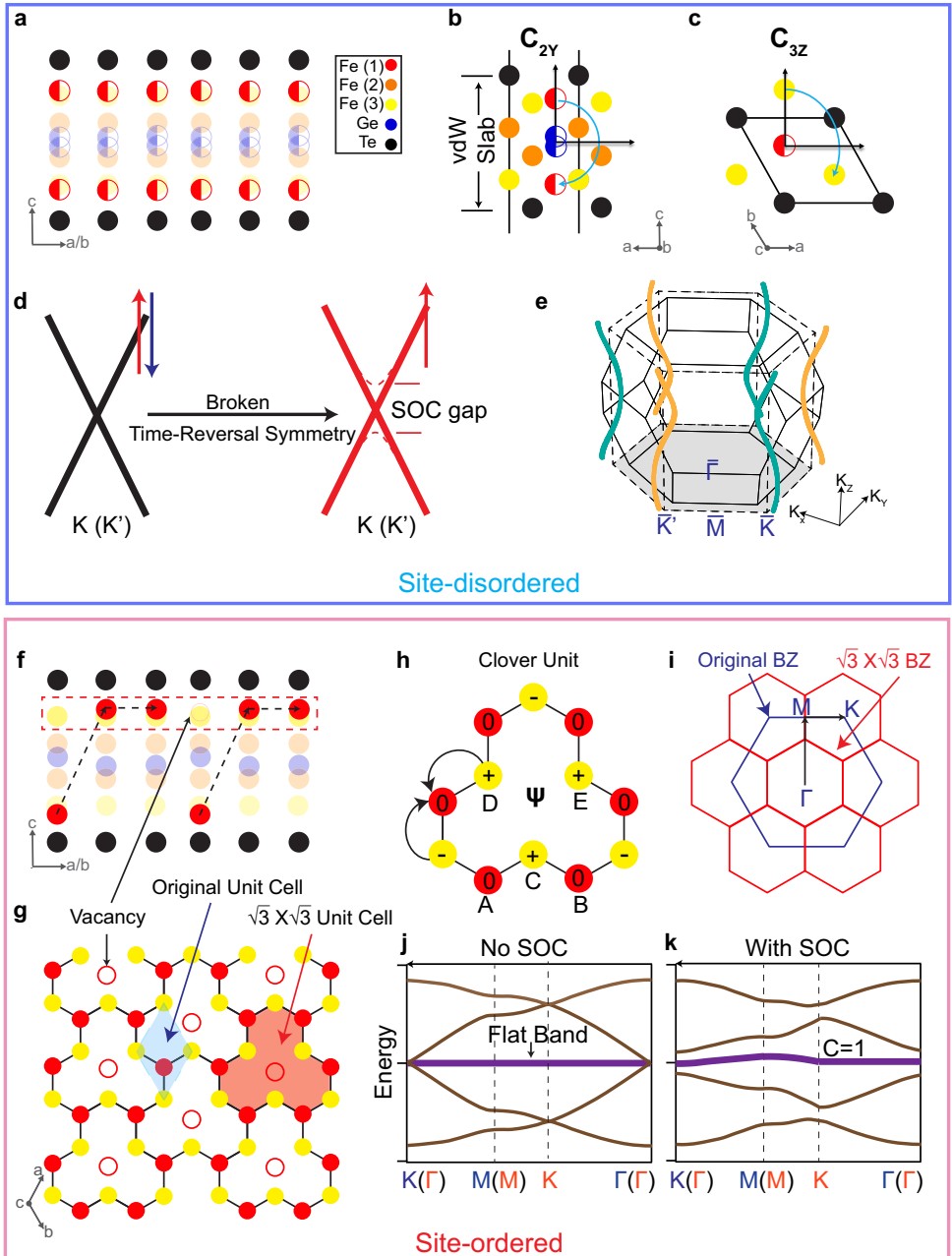

**Fig. 4 | Effect of the Fe(1) site ordering on inversion-symmetry and quantum destructive interference. a** Illustration of one vdW slab of the Fe(1) site-disordered phase, in which the Fe(1) sites are randomly distributed such that global inversion symmetry is preserved. **b, c** The schematic view of the lattice showing the $C_{2y}$ and $C_{3z}$ symmetries in the Fe(1) site-disordered phase, respectively. **d** The tight-binding model showing Dirac crossings at K (K′) protected by the above symmetries and the associated twofold topological crossing due to the ferromagnetic order and the gap by SOC. **e** The helical topological nodal lines induced by the ABC stacking of the vdW slabs that results in winding of the nodal lines with opposite chiralities around $\bar{K}$ and $\bar{K}'$. **f** Illustration of one vdW slab of the Fe(1) ordered-site

phase, in which the Fe(1) is DUU-ordered, forming a bipartite crystalline lattice. **g** The Fe(1) and the nearest neighbor Fe(3) sites form a clover lattice as shown in the red dashed box in (**f**). **h** Single clover unit showing the destructive interference of the hopping amplitude at the Fe(1) and Fe(3) sites due to the alternating sign of the Wannier phase, leading to a localization of the electronic wavefunction. The sites and wavefunction amplitudes are labeled on the corresponding atoms. **i** The original (blue) and the $\sqrt{3} \times \sqrt{3}$ superstructure (red) BZ. **j, k** Tight-binding model for the clover lattice without and with spin-orbital coupling (SOC), respectively. With SOC, the flat band is gapped with a Chern number of 1, and is hence topologically nontrivial.

Fig. 4d. This is consistent with our experimental observation of the crossing at K points except that the gap due to SOC is not resolved in the experiment due to the energy resolution. The appearance of this degeneracy at K points is similar to that reported in the related Fe$_3$GeTe$_2$, where the topological nodal lines are theoretically identified to give rise to a large anomalous Hall effect[57], but difficult to resolve in the ARPES measured dispersions. Here in Fe$_5$GeTe$_2$, they are clearly observed.

Having understood the single-layer case, we now consider the bulk system of the site-disordered phase. In a simple hypothetical AAA stacking scenario, the hopping along the z direction extends the original 2D hexagonal BZ into a 3D hexagonal prism and would extend the topological crossings at K and K′ to nodal lines along the K–H direction. This is protected by a combination of $C_{3z}$ and PT symmetries. For the real ABC stacking of the layers, the BZ changes from a hexagonal prism into the BZ of a rhombohedral space group (Fig. 4e). Due to the

ABC stacking of the layers, the K–H direction is no longer a high symmetry line of the BZ. Instead, the topological crossings at each $k_z$ plane shift away from the K and K′ points, forming helical nodal lines that wind around K–H, where the magnitude of the shift is proportional to the strength of the interlayer hopping, similar to helical nodal lines reported in other ABC-stacked materials including $Fe_3Sn_2$[58–60]. Here in $Fe_5GeTe_2$, due to the weak vdW interlayer coupling, the in-plane deviation of the crossing from the K–H line is too small to be experimentally resolved. Hence we cannot directly observe the winding but only observe nodal lines near K–H.

In addition to tight-binding calculations, we also carried out density functional theory (DFT) calculations to check for the symmetry-enforced crossings (see Supplementary Fig. 4a–c). To demonstrate the importance of the globally preserved inversion symmetry of the random Fe(1) occupation, we carried out the following comparison. First, we calculated the band structure for the Fe(1) sites all occupying the up sites (UUU). The inverted case is the structure with Fe(1) sites all occupying the down sites (DDD). The average of the two from directly overlapping the UUU and DDD band structures would give an average stoichiometry of $Fe_5GeTe_2$. We note that while such a structure does not exist in the crystal, it mimics the site-disordered phase except it lacks inversion symmetry. To directly compare this calculation with an inversion symmetric structure, we also calculated the band structure of a crystal structure with both up and down sites fully occupied, giving a stoichiometry of $Fe_6GeTe_2$. By comparing calculations without and with SOC, only the inversion symmetric $Fe_6GeTe_2$ shows band crossings at the K point that open up a gap with the inclusion of SOC, demonstrating the symmetry-enforced nature of the topological nodal lines. The UUU and DDD band structures do not exhibit such kind of band crossing, confirming that the presence of global inversion symmetry is consistent and also required for the observed topological nodal lines.

Finally, we discuss the site-ordered phase. Consistent with the inversion symmetry breaking observed by SHG, we no longer observe the topological crossings at the K point. Such inversion symmetry breaking is consistent with the $\sqrt{3} \times \sqrt{3}$ order caused by the Fe(1) site ordering. Interestingly, for such DUU occupation order of the Fe(1) site (Fig. 4f), the shortest bond occurs between the Fe(1) sublayer and the adjacent Fe(3) sublayer[47,48]. The in-plane projection of these two sublayers forms a clover unit pattern, with the center being the missing Fe(1) site, as shown in Fig. 4g, h. Considering only the nearest neighbor hopping, $t_1$, which is between the red Fe(1) sites and yellow Fe(3) sites, the lattice is manifested as a BCL, in which the lattice is categorized into two sublattices with different numbers of atoms (Fig. 4h). BCLs are predicted to be a generic platform to realize destructive interference of the electronic wavefunction and further lead to flat bands[61,62], but have never been directly observed in bulk materials.

To see this clearly, we consider the Hamiltonian for the single orbital clover lattice with nearest neighbor hopping

$$H(\boldsymbol{k}) = \begin{pmatrix} \mathbf{0}_{2\times2} & \mathcal{H}_{\boldsymbol{k}} \\ \mathcal{H}_{\boldsymbol{k}} & \mathbf{0}_{3\times3} \end{pmatrix}, \tag{3}$$

where $\mathcal{H}_{\boldsymbol{k}}$ is the hopping matrix between two sublattices (see explicit form in Supplementary Equation (17) in Supplementary Note 10). Given that $\mathcal{H}_{\boldsymbol{k}}$ is a $3 \times 2$ rectangular matrix, the Hamiltonian contains at least $3 - 2 = 1$ zero modes for all $\boldsymbol{k}$. The band structure is shown in Fig. 4j, after diagonalizing the Hamiltonian. The corresponding localized wavefunction for the flat band is:

$$\psi(k_x, k_y) = \begin{pmatrix} 0 & 0 & e^{\frac{k_y}{3}} - e^{-\frac{2}{3}k_y} & e^{-\frac{k_x}{2\sqrt{3}} - \frac{k_y}{6}} - e^{-\frac{k_x}{\sqrt{3}} + \frac{k_y}{3}} & e^{\frac{k_x}{2\sqrt{3}} - \frac{k_y}{6}} - e^{-\frac{k_x}{\sqrt{3}} + \frac{k_y}{3}} \end{pmatrix},$$
$$(4)$$

leading to a real space Wannier function as shown in Fig. 4h. The Wannier amplitude is identically zero on the red sites because of the destructive interference effect. To see this clearly, we have constructed an effective tight-binding model for the clover lattice (see a full description in the SI) that demonstrates that the Wannier amplitude is identically zero on the red sites, leading to a flat band (Fig. 4j). When SOC is incorporated, the flat band gains dispersion and also acquires a finite Chern number, and becomes topologically non-trivial. The consideration for different orbital groups is also provided in Supplementary Note 10. Such kind of destructive-interference-induced flat bands have been discussed in kagome[61–65] and pyrochlore[61,62] lattices. Here in quenched $Fe_5GeTe_2$, they are directly the result of the geometrically frustrated lattice formed by the Fe(1) occupation site ordering enabled by the quenching process.

We carried out DFT calculations to check for the BCL flat bands associated with the clover lattice (see Supplementary Fig. 4d). To mimic the site-disordered phase, we overlap the band structures for the UUU and DDD structures and compare it to the band structure calculated with the site-ordered phase with the $\sqrt{3} \times \sqrt{3}$ order. A direct comparison of the two shows that no flat bands are observed in the UUU + DDD calculation but flat bands are observed for the site-ordered phase. We can also unfold the band structure of the $\sqrt{3} \times \sqrt{3}$ order back to the original unfolded BZ to compare more directly with the observed dispersions, and find reasonable agreement (see Supplementary Fig. 5). Projection of the density of states unto the different Fe sites also shows that the peaks corresponding to the flat bands have large contributions from the Fe(1) and Fe(3) sites that form the clover lattice. This demonstrates that the flat dispersions that we observe only in the site-ordered phase and not in the site-disordered phase are associated with the clover unit that only forms with the Fe(1) site order, and is a direct result of the quantum destructive interference of the BCL.

## Discussion

Taking all experimental and theoretical evidence presented together, we have demonstrated the reversible switching of two remarkably distinct electronic structures ascribed to two closely related crystalline phases via a non-volatile thermal process in $Fe_5GeTe_2$. The capability is enabled by the Fe(1) site ordering that changes the crystal symmetries leading to distinct topological characteristics. On one hand, the random occupation of the Fe(1) sites leads to global inversion symmetry that allows symmetry-enforced topological nodal lines, which are observed to be lifted when the inversion-symmetry breaking order forms. On the other hand, the formation of the site order creates a BCL that localizes electronic states to form flat bands, which are observed to be destroyed with the breaking of the site order. Our findings indicate that the $Fe_5GeTe_2$ system is a rich system for probing and understanding topology in the correlated regime. As $Fe_5GeTe_2$ is known to exhibit high Curie temperature, it would be interesting to compare the magnetic properties of the two phases, including manipulation of the topological nodal lines in the site-disordered phase and the role of the topological flat bands for magnetism in the site-ordered phase. $Fe_5GeTe_2$ is also an interesting system to probe from the order-disorder perspective. We note that the quenching or slow cooling process produces samples that still exhibit both types of domains, but very different domain populations. We have encountered one case where the ARPES spectra exhibit an overlap of the two types of electronic structures (Supplementary Fig. 7) indicating a roughly equal population of the domains. This also explains the electrical transport properties of the samples produced in the two different ways being similar, as they likely are shorted by domain arrangements. As our STM results show that the slow-cooled samples exhibit domains of ordered regions, it would be interesting to understand how the domains form and propagate in the cooling process as a function of cooling rate, especially given that $Fe_5GeTe_2$ behaves

counterintuitively in that the ordered phase is preferred via quenching. Such studies would benefit from the vast expertize developed for probing and understanding order-disorder formation in other quantum materials[1–14]. Finally, the non-volatile switch our work exemplifies promises versatile settings to apply a design principle, viz. to utilize the cooperation of crystalline symmetry and strong correlations to produce various correlated topological materials[66].

Aside from fundamental physics, our work also indicates that $Fe_5GeTe_2$ has great potential for applications. Skyrmions have recently been reported in this class of Fe-based vdW ferromagnets[67–72]. As skyrmions are stabilized by Dzyaloshinskii–Moriya interaction, which is only allowed when inversion symmetry is broken, there have been debates on how to understand the appearance of skyrmions in these seemingly centrosymmetric crystals. In the case of $Fe_3GeTe_2$, this has been explained via random Fe deficiencies that on average occur asymmetrically in the crystal[73]. In the case of $(Fe,Co)_5GeTe_2$, this is ascribed to AA′ stacking of the vdW layers[74]. Here we show that the two phases have a clean distinction on inversion symmetry via the site-ordering process, hence providing a platform to potentially control skyrmion formation. The process by which we demonstrate the switching, heating and cooling all above room temperature, is similar to that already commercially used for phase-change materials such as $Ge_2Sb_2Te_5$[1,3,9]. Different from phase-change materials, we only need to surpass a submelting temperature where the Fe(1) sites are mobilized instead of having to achieve the melting and crystallization temperature. Techniques such as local laser heating can be explored for spatial writing of the two phases especially given that the overall crystal structures are compatible. This is in contrast to some vacancy-ordered materials such as $K_xFe_{2-y}Se_2$ where the metallic regions are structurally unstable and only appear as microstructures amidst the insulating vacancy-ordered phases[75,76]. The heating and quenching process that we utilize is non-volatile and above room temperature, which is advantageous compared to those controls that require the presence of field, strain, pressure, or current. Nevertheless, modifying the Fe(1) sites and their vacancies appears to have a lower energy barrier than re-crystallization, suggesting that electrical current, photo illumination or other commonly utilized switching methodologies could also be explored for this 2D vdW material.

Finally, the concept of using vacancy order-disorder to realize distinct topological phases goes beyond $Fe_5GeTe_2$. Phase change via order–disorder has been explored extensively for realizing switches based on electrical or optical properties[77]. Here we demonstrate the concept that vacancy order can be utilized to change the crystalline symmetries of two otherwise energetically similar ground states with dramatically distinct consequences on their topological character. A large base of quantum materials is known to exhibit vacancies or site disorder. The consideration of the symmetries of these phases may open up various routes towards realizing exotic topological phases as well as spintronics applications.

## Methods
### Crystal synthesis
Single crystals were grown via iodine-assisted chemical vapor transport following previous methods[48]. Fe powder, Ge pieces, and Te shot were weighed in the molar ratios 5:1:2, mixed, and placed within a quartz tube along with $2.5 \, mg/cm^3$ of $I_2$ pieces. The tube was then sealed under a low-pressure Ar atmosphere and the sealed tubes were placed in a horizontal furnace with one end open to air to create a natural temperature gradient with the source material at the center of the furnace. The furnace was ramped to 750 °C over 12 h, dwelled for 2 weeks, and then allowed to slowly cool back to room temperature for slow-cooled samples. $Fe_{5-\delta}GeTe_2$ single crystals with plate-like morphology and mirror-like surfaces after cleaving grew at the cold end of the quartz tubes. The crystals always exhibit finite Fe deficiency. Single crystal XRD refinements give a typical Fe deficiency of $\delta = -0.2$. As is

known, for quenched $Fe_{5-\delta}GeTe_2$ crystals, there is an irreversible phase transition near 100 K upon the first cool-down[48]. To avoid complications, all our measurements presented for all techniques start after the first cool-down. For re-quenching a crystal, we sealed the crystals in a quartz ampoule, slowly ramped to 750 K, and annealed for 2 h then quenched in cold water.

### Electrical transport and magnetization
Samples were cleaved and shaped into a bar morphology with a scalpel. Gold contacts in a six-point geometry were sputtered onto the sample using a paper mask, and electrical leads were attached in the form of gold wire using H20E Epo-Tek silver epoxy (Ted Pella, Inc). Electrical resistivity was measured via a lock-in amplifier in a Quantum Design PPMS Dynacool using a constant current source of $I = 1 \, mA$. For magnetic field-dependent measurements, $\rho_{xx}$ and $\rho_{xy}$ were symmetrized and anti-symmetrized, respectively for positive and negative fields. Magnetization measurements were performed using the VSM option of the Quantum Design PPMS Dynacool. Samples were cleaved and shaped using a scalpel and mounted onto sample holders with a small amount of GE varnish.

### TEM characterization
To check the stacking order for slow-cooled and quenched crystals, we carried out TEM measurements to probe and compare the cross-sections. The quenched crystal was from the same piece of re-quenched crystal post-ARPES measurement (Fig. 3). The crystal was prepared from a slow-cooled crystal that we annealed and quenched post-growth. HAADF-STEM imaging was acquired on an aberration-corrected TEM (FEI, TITAN) at 300 kV. A 25 mrad convergence angle and a 40 mrad inner collection angle is used. The contrast of HAADF images is proportional to $Z^\gamma$, where $Z$ is the atomic number and $1.3 < \gamma < 2$. Cross-sectional TEM samples were cut on a dual-beam FIB/SEM (FEI Helios 660), with an ending voltage at 2 keV of the ion beam for final thinning. The image was taken along [100] direction on a slow-cooled sample which was reported to have more stacking fault comparing those quenched sample[47,48]. The details can be found in Supplementary Note 1.

### SHG measurements
Bulk flakes of $Fe_{5-\delta}GeTe_2$ samples with crystal $ab$-plane orientation were exfoliated onto commercial $SiO_2$/Si wafers, which were loaded into a closed-cycle cryostat. An 800 nm Ti:Sapphire laser (80MHz, ~50 fs) used as the fundamental light is focused to a 2 µm spot onto the sample by a 50× objective. All measurements were done at normal incidence. A typical laser power of ~4 mW was used for SHG measurements on these bulk crystals. No obvious sample degradation was observed during the measurements. The same objective was applied to pick up the output 400 nm SHG beam, which was then guided by a dichroic mirror to a photon counter. A half-waveplate and a linear polarizer were used to control the polarization of incident fundamental light and output second-harmonic light, respectively. Via co-rotating the half waveplate and linear polarizer, both SHG polar patterns with crossed and parallel configurations are measured. We measured two types of crystals at 5 K, those that were slowly cooled in the furnace (slow-cooled) and those that were initially slow-cooled and then re-quenched post-growth. The details can be found in Supplementary Note 2.

### Single-crystal diffraction
Hard single-crystal XRD is carried out at the energy of 29 KeV at beamline QM2 of the Cornell High Energy Synchrotron Source (CHESS). The sample is mounted with GE Varnish on a rotating pin before being placed in the beam. A 6-MP hybrid photon counting detector (Pilatus3 6M detector) is used to collect the diffraction pattern with the sample rotated 365° around three different axes at 0.1°

step and 0.1s/frame data rate at the measurement temperature of 300 K. The full 3D intensity cube is stacked and indexed with beamline software. The XRD analysis can be found in Supplementary Note 3.

## STM measurements

We carried out STM measurements on the two types of samples: one is slow-cooled from the furnace, while the other one is re-quenched from a sample grown in the same batch. All STM measurements were taken with a Unisoku USM1300 at 4.2 K using a PtIr tip. The samples were cleaved in situ under ultra-high vacuum conditions. Bias spectra (dI/dV) were taken using a lock-in technique with a modulation amplitude of 5 mV. More information is available in Supplementary Note 4.

## ARPES measurements

ARPES measurements were carried out at beamline 5-2 of the Stanford Synchrotron Radiation Lightsource, ESM (21ID-I) beamline of the National Synchrotron Light Source II, and beamlines 10.0.1 and 4.0.3 of the Advanced Light Source, using a DA30, DA30, R4000, and R8000 electron analyzer, respectively. The energy and angular resolutions were set to 20 meV and 0.3°, respectively. The samples were cleaved in situ at base temperature (between 15 and 20 K) and kept in an ultra-high vacuum with a base pressure lower than $5 \times 10^{-11}$ Torr during measurements. All measurements presented were performed at base temperature in the ferromagnetic phase. The definition of the light polarizations LH and LV are as illustrated in Supplementary Fig. 12. For each type of crystal (quenched, slow-cooled, and re-quenched), we have checked and measured 5–20 crystals each, all showing consistent behavior, namely the observation of either the topological crossing type or the flat band type. All with the exception of one case where we observed an overlap of the two types of electronic phases. This is discussed in the Supplementary Notes 5–7.

## DFT calculations

We carried out DFT calculations by using the Vienna ab initio simulation package (VASP)[78], where the plane-wave cut-off was set to 500 eV. We used Perdew–Burke–Ernzefhof (PBE) generalized gradient approximation (GGA) for the exchange–correlation functional, and a $10 \times 10 \times 2$ K point mesh for self-consistent calculation[79]. Detailed discussion is available in Supplementary Note 8. The major source of error in DFT calculations is its inability to accurately capture exchange and dynamical correlation, which may be crucial in Fe-based compounds. Further investigation and discussion is awaiting.

## Data availability

The authors declare that the data supporting the findings of this study are available within this article and the Supplementary Information file. The source data are also available from the corresponding author upon request.

## Code availability

Results can be reproduced using standard VASP packages. Methods are fully described. Codes used to produce figures can be made available upon request.

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

## Acknowledgements

The authors acknowledge insightful discussions with Luis Balicas, Alex Frano, Andrew May, and Kai Sun. This research used resources of the Advanced Light Source, the Stanford Synchrotron Radiation Lightsource, and the National Synchrotron Light Source-II, all U.S. Department Of Energy (DOE) Office of Science User Facilities under contract Nos. DE-AC02-05CH11231, DE-AC02-76SF00515, and No. DE-SC0012704, respectively. Rice ARPES work is supported by the U.S. DOE, Office of Basic Energy Sciences (BES), under Award No. DE-SC0021421, the Gordon and Betty Moore Foundation's EPiQS Initiative through Grant no. GBMF9470, and the Robert A. Welch Foundation (Grant No. C-2175 to M.Y.). The theory work at Rice is primarily supported by the U.S. DOE, BES, under Award No. DE-SC0018197 (L.C., symmetry

analysis), by the Air Force Office of Scientific Research (AFOSR) under Grant No. FA9550-21-1-0356 (C.S., electronic structure construction), and by the Robert A. Welch Foundation Grant No. C-1411 and the Vannevar Bush Faculty Fellowship ONR-VB N00014-23-1-2870 (Q.S.). Work at Los Alamos was carried out under the auspices of the U.S. DOE National Nuclear Security Administration (NNSA) under Contract No. 89233218CNA000001 and was supported by LANL LDRD Program, UC Laboratory Fees Research Program (Grant No. FR-20-653926), and in part by the Center for Integrated Nanotechnologies, a DOE BES user facility. The development of the SHG photon counter is supported by the Army Research Office and was accomplished under Grant no. W911NF-19-1-0342. The sample exfoliation is based upon work supported by the AFOSR under award number FA9550-22-1-0449. Q.D. is supported by the National Science Foundation (NSF) EPM program under Grant no. DMR-2213891. L.W. acknowledges the support by the AFOSR under award no. FA9550-22-1-0410. TEM study is supported by Welch Foundation (C-2065-20210327). The authors acknowledge the use of the Electron Microscopy Center at Rice. The work at LBL and UC Berkeley was funded by the U.S. DOE, Office of Science, BES, Materials Sciences and Engineering Division under Contract No. DE-AC02-05-CH11231 (Quantum Materials Program KC2202). Research conducted at the Center for High-Energy X-ray Sciences (CHEXS) is supported by the NSF (BIO, ENG, and MPS Directorates) under award DMR-1829070. Materials synthesis at UW was supported as part of Programmable Quantum Materials, an Energy Frontier Research Center funded by the U.S. DOE, Office of Science, BES, under award DE-SC0019443. Single crystal annealing process at Rice was supported by NSF Grant No. DMR-2100741 and by the Robert A. Welch Foundation under Grant No. C-1839 (P.D.). E.H.d.S.N. acknowledges support by the NSF under Grant No. DMR-2034345 and the Alfred P. Sloan Research Fellowship. M.H. and D.L. acknowledge the support of the U.S. DOE, Office of Science, BES, Division of Material Sciences and Engineering, under contract DE-AC02-76SF00515.

## Author contributions

The project was initiated and organized by M.Y. The single crystals were grown by P.M., Y.S., X.C., C.H., X.X., and J.-H.C. The ARPES measurements and analyses were carried out by H.W., J.H., Z.Y., J.S.O., Y.G., T.W., Y. He, J.K., R.J.B., and M.Y. with the help of D.L., M.H., S.-K.M., A.F., J.D.D., T.Y., and E.V. The tight binding model and symmetry analysis were proposed and carried out by L.C., C. Setty, and Q.S. The first principle calculations were carried out by B.G.J. and J.-X.Z. The SHG was carried out by Q.D. and L.W. The STM measurements were measured by K.S. and E.H.d.S.N. The XRD was done by J.P.C.R. The TEM measurements were carried out by C. Shi and Y. Han. The sample annealing and quenching process and characterization were carried out by P.M., J.-H.C., X.C., Y.X., B.G., X.T., M.K., H.W., and P.D. The manuscript was written by H.W. and M.Y. and contributed by all the authors.

## Competing interests

The authors declare no competing interests.

## Additional information

[1]Department of Physics and Astronomy and Rice Center for Quantum Materials, Rice University, Houston, TX, USA. [2]Department of Physics, University of Washington, Seattle, WA, USA. [3]Theoretical Division and Center for Integrated Nanotechnologies, Los Alamos National Laboratory, Los Alamos, NM, USA. [4]Department of Advanced Materials Engineering for Information and Electronics, Kyung Hee University, Yongin, Republic of Korea. [5]Department of Physics and Astronomy, University of Pennsylvania, Philadelphia, PA, USA. [6]Department of Physics, Yale University, New Haven, CT, USA. [7]Energy Sciences Institute, Yale University, West Haven, CT, USA. [8]Department of Physics and Astronomy, University of California, Davis, CA, USA. [9]Department of Applied Physics, Yale University, New Haven, CT, USA. [10]Cornell High Energy Synchrotron Source, Cornell University, Ithaca, NY, USA. [11]Department of Physics, University of California, Berkeley, CA, USA. [12]Department of Materials Science and Engineering, University of Washington, Seattle, WA, USA. [13]Department of Materials Science and NanoEngineering, Rice University, Houston, TX, USA. [14]Stanford Synchrotron Radiation Lightsource, SLAC National Accelerator Laboratory, Menlo Park, CA, USA. [15]National Synchrotron Light Source II, Brookhaven National Lab, Upton, NY, USA. [16]Advanced Light Source, Lawrence Berkeley National Laboratory, Berkeley, CA, USA. [17]Departments of Electrical and Computer Engineering, Rice University, Houston, TX, USA. [18]Materials Sciences Division, Lawrence Berkeley National Laboratory, Berkeley, CA, USA. [19]Department of Materials Science and Engineering, University of California, Berkeley, CA, USA. ✉e-mail: mingyi@rice.edu

