## [Peer Review File · Nature Communications]

Reversible non-volatile electronic switching in a near-room-temperature van der Waals ferromagnetEditorial Note: This manuscript has been previously reviewed at another journal that is not operating a transparent peer review scheme. This document only contains reviewer comments and rebuttal letters for versions considered at Nature Communications.

Reviewers' Comments:

Reviewer #1:

Remarks to the Author:

I have read the revised manuscript and the response to referees. My main criticisms upon the last review regarded the case for the impact of the manuscript. The manuscript has not been edited in response to my previous comments as far as I can see, so there is little for me to say.

I am partially swayed by the arguments in the rebuttal letter for the prospects of the field moving forward, and I expect that the paper will have some impact. I think the scientific aspects are well presented and I have no further comments. The paper should be published in some venue. If the editor feels Nature Comms is appropriate given the other reports, I do not object.

Reviewer #2:

Remarks to the Author:

In this manuscript, the authors report a non-volatile method to reversibly toggle between two distinct states in van der Waals ferromagnet Fe_5GeTe_2 , the experimental data are convincing and interesting, especially, they directly observed the topological nodal line in a crystalline system with disorder in the site-disordered phases for the first time. I think it is acceptable to publish the results in Nature Communication.